# Efficacy and long-term safety of CRISPR/Cas9 genome editing in the *SOD1*-linked mouse models of ALS

Han-Xiang Deng [1✉], Hong Zhai[1], Yong Shi[1], Guoxiang Liu[1], Jessica Lowry[1], Bin Liu[1], Éanna B. Ryan[1], Jianhua Yan[1], Yi Yang[1], Nigel Zhang[1], Zhihua Yang[1], Erdong Liu[1], Yongchao C. Ma [2] & Teepu Siddique [1✉]

CRISPR/Cas9-mediated genome editing provides potential for therapeutic development. Efficacy and long-term safety represent major concerns that remain to be adequately addressed in preclinical studies. Here we show that CRISPR/Cas9-mediated genome editing in two distinct *SOD1*-amyotrophic lateral sclerosis (ALS) transgenic mouse models prevented the development of ALS-like disease and pathology. The disease-linked transgene was effectively edited, with rare off-target editing events. We observed frequent large DNA deletions, ranging from a few hundred to several thousand base pairs. We determined that these large deletions were mediated by proximate identical sequences in *Alu* elements. No evidence of other diseases was observed beyond 2 years of age in these genome edited mice. Our data provide preclinical evidence of the efficacy and long-term safety of the CRISPR/ Cas9 therapeutic approach. Moreover, the molecular mechanism of proximate identical sequences-mediated recombination provides mechanistic information to optimize therapeutic targeting design, and to avoid or minimize unintended and potentially deleterious recombination events.

[1] The Ken and Ruth Davee Department of Neurology, Feinberg School of Medicine, Northwestern University, Chicago, IL, USA. [2] Departments of Pediatrics, Neurology and Physiology, Ann & Robert H. Lurie Children's Hospital of Chicago, Feinberg School of Medicine, Northwestern University, Chicago, IL, USA. ✉email: h-deng@northwestern.edu; t-siddique@northwestern.edu

Amyotrophic lateral sclerosis (ALS) is a uniformly fatal disease caused by degeneration of motor neurons in the central nervous system, leading to respiratory failure and death, typically within 3 years of symptom onset. Currently, no effective treatment is available for ALS. Mutations of *SOD1* are associated with ~20% of familial ALS[1,2]. Patients with an SOD1-A4V mutation, which accounts for approximately half of all *SOD1*-linked ALS cases in North America, have a rapid progression of disease, with a mean disease duration of only $1.0 \pm 0.4$ years[3], highlighting the urgency of the unmet need for therapeutic development. Previous studies have shown that transgenic mice overexpressing ALS-linked mutant human *SOD1* (*hSOD1*) develop an ALS-like phenotype and pathology[4,5], which were absent in *Sod1* deficient mice[6]. These data provide a rational basis for genetic targeting/editing of *SOD1* as a therapeutic strategy, which may prove to be a "once for all" solution to treat *SOD1*-linked ALS.

CRISPR/Cas9-mediated genome editing is a powerful tool that can be used to target specific DNA sequences in vitro and in vivo[7–9]. This technology provides great potential for therapeutic development, but efficacy and long-term safety remain primary concerns in its clinical application. Off-target editing events, large DNA deletions, and other rearrangements are major safety concerns[10–13], which remain to be adequately addressed in preclinical studies in vivo.

Previous studies have shown that in vivo genome editing of disease-linked genes by CRISPR/Cas9 can ameliorate disease phenotype and pathology in animal models without inducing severe adverse effects[14–23]. These data provide proof-of-concept evidence that CRISPR/Cas9-mediated genome editing has potential therapeutic applications. However, several factors may preclude detection of long-term adverse effects, including low editing efficiency, a limited number of edited cells, and a relatively brief period of post editing monitoring. One major long-term safety concern is tumorigenesis that might arise if off-targets involve oncogenes and tumor suppressor genes. It is known that tumor development may be delayed for a certain period of time after the initial insult. For example, it took half a year for the mice with a germline homozygous *p53* deletion to develop lymphoma and sarcoma[24]. Also, it took one and a half years for mice with a liver-specific homozygous *p53* deletion to develop hepatocellular carcinoma[25]. Most previous studies reported their safety observations for a few weeks or months after introduction of CRISPR/Cas9. A few studies reported observations 13 and 19 months after introduction of CRISPR/Cas9[20,22]; however, low targeting/editing efficacy might mask the potential long-term risks.

Another major safety concern is CRISPR/Cas9-mediated large DNA deletions at target sites[11,12]. These large deletions can range from several hundred to several thousand base pairs (bp)[11,12]. The mechanism underlying the emergence of these large deletions remains to be determined. Understanding the molecular mechanism underlying the emergence of these large deletions may provide mechanistic information to help avoid or minimize the risk of such events.

The goal of the present study is to address the long-term safety concerns in a situation where a maximum editing efficiency is achieved in an in vivo disease model. We chose human SOD1-G93A transgenic mouse models of ALS for this purpose, as these mouse models display a highly predictable disease course, pathology, and narrow windows of disease onset and survival[4,5]. We designed a transgenic strategy to edit the disease-linked gene by the expression of a specific single guide RNA (gRNA) and CRISPR/Cas9 from an early embryonic stage, with the gRNA and CRISPR/Cas9 expression persisting throughout the lifespan of the mice. With this approach, we expected not only maximum editing efficiency and maximum therapeutic efficacy, but also maximum incidence of adverse effects due to efficient genome editing of every somatic cell during the lifespan of the mice. Here, we report that CRISPR/Cas9-mediated editing of the ALS-linked human SOD1-G93A transgenes (*hSOD1-G93A*) in two transgenic models completely prevented the development of any signs of clinical ALS or ALS pathology. Transgene-edited mice appeared healthy with no overt signs of other diseases, such as tumorigenesis or inflammatory diseases, even beyond 2 years of age. We demonstrate that all copies of the transgene were effectively edited, with limited off-target editing events. We also report the unexpected finding of a high incidence of large DNA deletions and the molecular mechanism underlying the emergence of these large deletions, providing mechanistic information that can be used to optimize therapeutic targeting design, and to avoid or minimize unintended and potentially deleterious events.

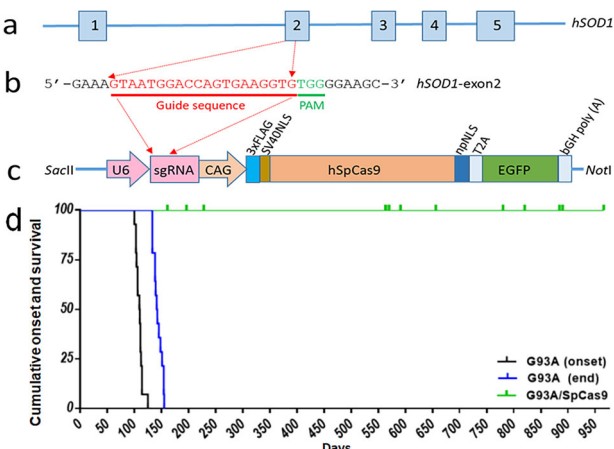

**Fig. 1 Schematic representation of the human SOD1-SpCas9 transgene. a** Human *SOD1* structure. **b** A 20 bp DNA sequence (in red) of *hSOD1*-exon2 was used to express guide RNA, and this 20 bp sequence was inserted into the BbSI site of a plasmid vector, pSpCas9 (BB)-2A-GFP (PX458) (**c**). The plasmid DNA was digested with *Sac*II and *Not*I. The 7.5 kb transgene was used for microinjection into mouse fertilized eggs. The protospacer adjacent motif (PAM) is shown in green (**b**). U6 U6 promoter, sgRNA single guide RNA containing a targeting sequence (crRNA sequence), and a Cas9 nuclease-recruiting sequence (tracrRNA), CAG a hybrid promoter[26], SV40NLS simian virus 40 nuclear localization signal, hSpCas9 a human codon optimized SpCas9, npNLS nucleoplasmin nuclear localization signal, T2A a viral 2A self-cleaving peptide, EGFP enhanced green fluorescent protein, bGH poly (A) bovine growth hormone poly (A) signal. Diagrams are not drawn to scale for clarity. **d** Kaplan–Meier plots showing the age of onset (black line, $109.5 \pm 6.3$ days) and cumulative survival (end stage, blue line, $141.5 \pm 8.1$ days) of G93A high expresser line (G1H, $n = 14$) and G1H/Cas9 double transgenic mice (green, $n = 15$). The G1H/Cas9 mice did not develop an ALS-like phenotype.

## Results

**Prevention of ALS-like phenotype by CRISPR/Cas9-mediated *hSOD1*-G93A transgene editing.** To maximize editing efficiency, we designed a transgenic approach to express an *hSOD1*-specific single gRNA and CRISPR/Cas9, which are expected to be expressed from the zygote stage throughout the entire lifespan of the mouse[26]. This strategy offers advantages to address long-term safety concerns in vivo (i.e., a minimum of 2 years). To specifically target *hSOD1*, we carefully selected a unique sequence of 20 nucleotides (nt) in exon2 of *hSOD1* as a template for gRNA synthesis (Fig. 1). We constructed a transgene by insertion of this 20 nt sequence into a plasmid vector, pSpCas9 (BB)-2A-GFP (PX458) (Fig. 1a–c). The plasmid DNA was digested with *Sac*II

and *Not*I, and the 7.5 kilobase (kb) transgene was used for microinjection into mouse fertilized eggs from a *C57BL/6J* inbred strain (Fig. 1c). We identified four *hSOD1*-Cas9 (Cas9) transgenic founders among 16 pups. Since safety is the primary concern that needs to be addressed when Cas9 is expressed at a high level for a prolonged period of time, we established a founder line (#2226) with the highest copy number for subsequent studies.

We crossbred the Cas9 transgenic mice with *hSOD1*-G93A transgenic mice that harbor a high transgene copy number (G1H)[4], to generate double transgenic mice (G1H/Cas9). Consistent with previous observations, disease onset in the control G1H single transgenic mice on the *C57BL/6J* genetic background occurred by 3–4 months, with a survival of 4–5 months[4]. However, all 15 G1H/Cas9 double transgenic mice remained phenotypically normal, even though they were all over 6 months old, with the oldest mice having lived for ~32 months (Fig. 1d).

We also tested Cas9 safety and efficacy in another *hSOD1*-G93A transgenic line with a reduced transgene copy number (G1L). Similar results were observed in two G1L/Cas9 double transgenic mice, which were phenotypically normal beyond 15 months, while control G1L mice had a survival of only ~8 months.

In our Cas9 transgene, gRNA and Cas9 expression are driven by the U6 and CAG promoter, respectively. Both the gRNA and Cas9 are constitutively expressed from fertilized eggs throughout the life of the mice[26]. To assess the long-term effects of gRNA and Cas9 expression on the mouse phenotype, we maintained and monitored Cas9 transgenic mice during their natural lifespan. The founder mouse (#2226) lived for 34 months without notable tumor development or inflammatory disease. We also monitored the phenotype of 36 Cas9 mice for a minimum of 2 years, and these mice appeared normal and there was no indication of tumorigenesis or inflammatory disease.

**Prevention of ALS-like pathology by CRISPR/Cas9-mediated *hSOD1*-G93A transgene editing.** The presence of skein-like inclusions in surviving motor neurons is a pathological feature in ALS. In *SOD1*-linked ALS patients and *SOD1*-G93A transgenic mice, these inclusions are immunoreactive to antibodies against SOD1, p-62, and ubiquitin[5]. In addition to these inclusions, mitochondrial vacuoles are also prominent[5]. Consistent with our previous findings[5], we observed protein inclusions, mitochondrial vacuoles, microglial activation, and astrocytosis in control G1H mice (Fig. 2a–c). However, these pathological changes were absent in the G1H/Cas9 mice (Fig. 2d–f). In addition, the shrinkage and loss of motor neurons in the anterior horn, loss of large-calibers axons in anterior roots, and muscle atrophy were observed in the control G1H mice, but not in the genome edited G1H/Cas9 mice (Fig. 2g–l, Supplementary Fig. 1). The absence of an ALS-like phenotype and pathology in the G1H/Cas9 mice demonstrates that the CRISPR/Cas9-mediated genome editing approach is highly effective.

**Complete editing of *hSOD1* by CRISPR/Cas9 transgenic approach.** To assess the editing efficiency of CRISPR/Cas9 at the transgene level as accurately as possible, we employed a deep sequencing approach. We initially amplified the genomic DNA from a G1H/Cas9 mouse (#8190) using a pair of primers specific to *hSOD1*, covering a 512 bp fragment flanking the Cas9 cleavage site. The primers were anchored with an *Eco*R1 or a *Hind*III site, respectively. The PCR products were cloned into a plasmid vector, *pBluescript* II SK(-), and individual clones were directly analyzed by Sanger sequencing to determine the precise editing events. We analyzed a total of 112 individual clones, and

identified 19 different editing events (Fig. 3a, Supplementary Fig. 2). No wild-type clones were present. We also analyzed a total of 117 individual clones derived from a G1L/Cas9 mouse (#8306), and identified eight different editing events (Supplementary Fig. 3). Again, no wild-type clones were found. These data suggest that all copies of the *hSOD1*-G93A transgene in both G1H/Cas9 and G1L/Cas9 mice were fully edited. Notably, some editing events in the transgene-edited mice accounted for a very small fraction (<1%) (Fig. 3a, Supplementary Fig. 3), suggesting that these events occurred in multiple cell stages in early embryonic development. These small insertions/deletions could be explained by the pathways of classical nonhomologous end joining (cNHEJ) (such as insG) and alternative end joining (Alt-EJ or microhomology-mediated end joining, MMEJ) (such as delGTGAAG) (Fig. 3 and Supplementary Fig. 3)[27–29].

To evaluate mutant human SOD1 protein expression after CRISPR/Cas9-mediated *hSOD1* editing, we performed western blot analysis using three SOD1 antibodies: antibody c-SOD1 recognizes both human and mouse SOD1 [5]. A human-specific SOD1 antibody (hs-SOD1) recognizes only human SOD1, but not mouse Sod1[5]. We also generated a mouse-specific Sod1 antibody (ms-Sod1), which only recognizes mouse Sod1, but not human SOD1. We did not detect the mutant human SOD1 in the genome edited G1H/Cas9 mice. As expected, the endogenous mouse Sod1 was not affected. Consistent with our pathological data, neuroinflammation was observed in G1H, but not in the G1H/Cas9 mice, as shown by immunoblots with antibodies against Aif1 (Iba1) and Gfap (Fig. 3b and Supplementary Fig. 4).

**Off-target editing events.** Off-target editing events are a primary safety concern for therapeutic genome editing in vivo. The rate of off-target editing events can be influenced by multiple factors, such as gRNA sequence specificity and structure, the location of mismatches in gRNA, and concentration of the gRNA and Cas9[8,10,30–32]. Initial studies in vitro suggested that the SpCas9 protospacer-adjacent motif (PAM) sequence "NGG" and the 12–13 base "seed sequence" at the 3′ end of the gRNA are critical for DNA cleavage specificity[7,8]. In addition, "NAG" may also be used as a PAM, albeit with a lower efficiency than "NGG"[31,33]. We carefully selected our sgRNA sequence so that there are at least two mismatches between the gRNA and any potential off-targets in mouse genome to minimize potential off-target events. We used a variety of silico off-target programs and a relaxed "NRG" PAM (including NGG and NAG) to predict potential off-targets.

We analyzed 15 potential off-targets in a transgene-edited G1H/Cas9 mouse (#8190), in which all *hSOD1* copies were fully edited, using DNA cloning and deep sequencing approaches. The primers for PCR amplification were anchored with an *Eco*RI or a *Hind*III site, respectively. The PCR products were cloned into a plasmid vector, *pBluescript* II SK(-). We deep-sequenced over a 100 individual plasmid clones for each of the 15 potential off-targets, so that the off-target editing events with a rate of over 1% at any single site might be detected. We identified three clones with different sequence alterations (insG, delAGinsGG, and delGTinsGAGTGGTCA) from a total of 132 clones derived from off-target #01 (Fig. 4a, c). No sequence alterations were observed in the other 14 potential off-targets (Fig. 4a). We further analyzed off-target editing events for off-targets #01 and #08 in another G1H/Cas9 mouse (#7449). Among 156 clones of off-target #01, we identified two clones with sequence variations (delG and delAAGG) (Fig. 4b, c). Again, we did not find any sequence variations among 185 clones for off-target #08 in this transgene-edited G1H/Cas9 mouse (Fig. 4b). These data suggest that mismatches in the gRNA, especially those near PAM are less

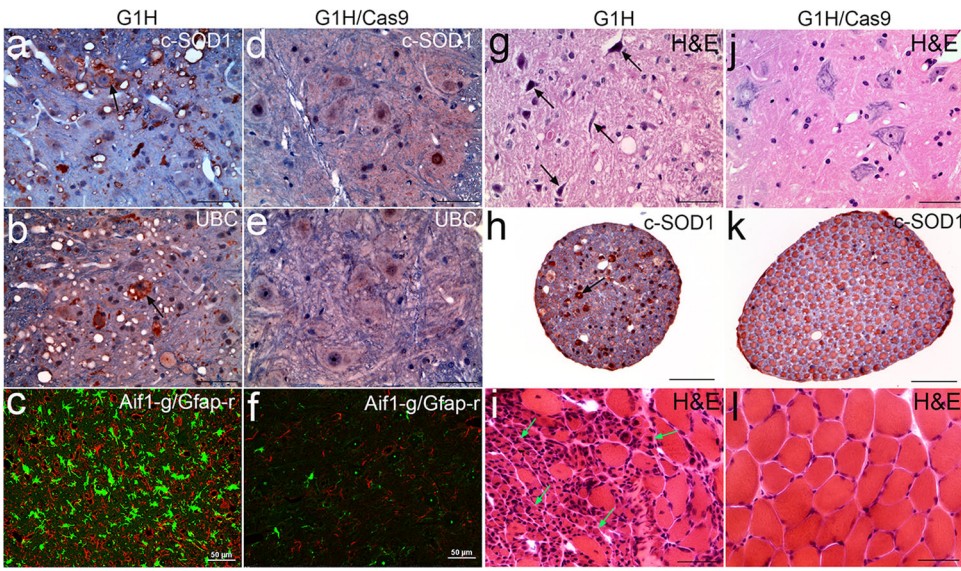

**Fig. 2 ALS-free pathology in G1H/Cas9 mice.** The spinal cord sections were stained and analyzed using immunohistochemistry (**a**, **b**, **d**, **e**) or confocal microscopy (**c**, **f**) using the indicated antibodies. Representative images from an end stage G1H (#3702, 148 days) and *hSOD1*-targetd G1H/Cas9 mice (#7450, 196 days) are shown. Immunoreactive aggregates are indicated by arrows (**a**, **b**). **c**, **f** Microglial activation and astrocytosis are shown by Aif1 (green) and Gfap (red) staining, respectively. **g**, **j** H&E staining of the spinal cord sections showing motor neuron shrinkage (arrows) in the anterior horn of G1H mice. **h**, **k** Loss of large-caliber axons in an anterior root in G1H mice. Arrow indicates SOD1 aggregates in a motor axon. **i**, **l** H&E staining of the gastrocnemius muscles showing muscle fiber atrophy (green arrows) in the G1H mice. Scale bar, 50 μm. These pathological changes are absent in the G1H/Cas9 mice.

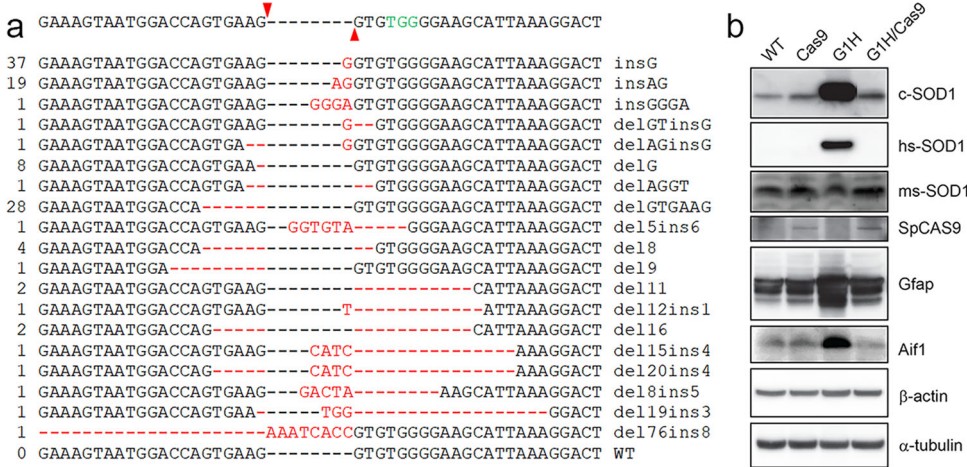

**Fig. 3 Efficient targeting of *hSOD1* in the G1H/Cas9 mice. a** Targeting events identified in a G1H/Cas9 mouse (#8190, 585 days). Among 112 individual clones analyzed, 19 different targeting events were identified. PAM sequence (TGG) is labeled in green. Red arrowheads indicate the Cas9 cleavage site. The deleted nucleotides are shown by red dashed lines. Red letters represent the inserted nucleotides. The number of clones harboring the indicated mutation is shown on the left. Individual mutations are on the right. For deletions exceeding six nucleotides, the deleted nucleotides are represented by numbers for clarity. **b** Efficient removal of *hSOD1* in the G1H/Cas9 mice. Immunoblotting of the spinal cord homogenates from mice was performed with antibodies indicated on the right. β-actin and α-tubulin were used as internal loading controls.

tolerated in vivo than in vitro, possibly due to much higher transient expression of gRNA and Cas9 in the in vitro system. Notably, #08 has one base more than #01 in the "seed sequence" near the PAM, but the #01 PAM is "GGG," while the #08 PAM is "AGG" (Fig. 4a, b). The apparent difference in the targeting rate between #01 and #08 in two independent mice raises the possibility that the PAM sequence "GGG" may be more effective than "AGG" in mediating CRISPR/Cas9 editing.

We analyzed off target events in two mice at ages of 196 days (#7449) and 585 days (#8190), respectively. The off-target rate appeared to increase with age in the CRISPR/Cas9 mice [1.28% (2/156) at 196 days vs 2.27% (3/156) at 585 days]. However, our

limited sample size ($n = 2$) and the total number of analyzed clones ($156 + 132 = 288$) might not be sufficient to draw a reliable conclusion. Future studies may be directed to analyze multiple mice at different ages using targeted PCR combined with next-generation sequencing strategies, so that a sufficient number of individual fragments could be characterized at different ages.

**Frequent large deletions and underlying mechanism.** In addition to small insertions/deletions in the immediate vicinity of the Cas9 cleavage sites, large deletions may also occur within these regions. This poses another major safety concern, as large

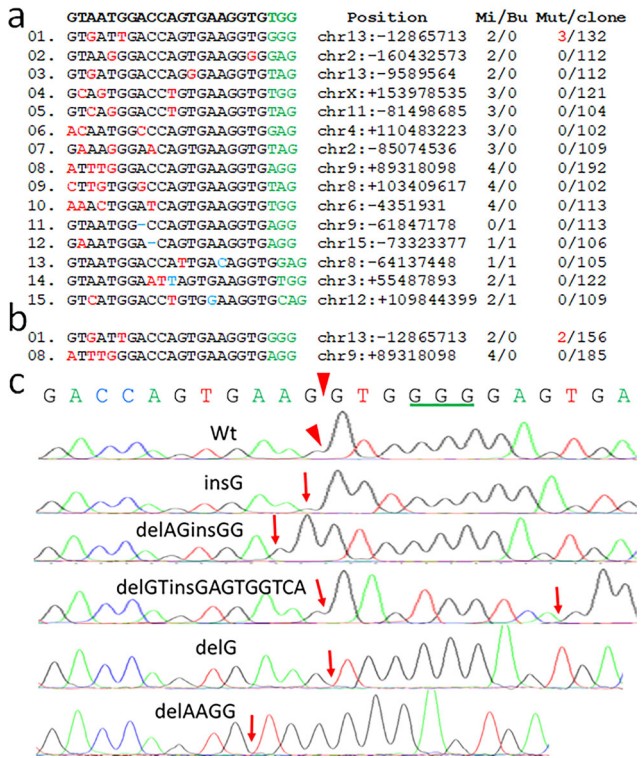

**Fig. 4 Off-target editing events in the G1H/Cas9 mice. a, b** *hSOD1* sequence for Cas9 gRNA is on the top. DNA sequence, chromosomal position, number of nucleotide mismatch and bulge (Mi/Bu), and number of the identified mutants among the total number of the sequenced clones (Mut/clone) in 15 genome-wide potential off-target sites are shown. PAM sequences are shown in green. Nucleotide mismatches are shown in red. Nucleotide in blue indicates DNA bulge and (−) in blue indicates RNA bulge. Two G1H/Cas9 mice were analyzed (#8190, 585 days: **a**; #7449, 196 days: **b**). **c** Editing events identified in off-target #1. Arrowheads indicate the Cas9 cleavage site in off-target #1. Individual edited events are labeled on the top of sequencing chromatograms. Among 132 clones, three mutant clones were identified (delG, delAGinsGG, and delGTinsGAGTGGTCA) in mouse #8190, and among 156 clones, two mutant clones were identified (delG and delAAGG) in mouse #7449. The sites of mutations are indicated by arrows.

deletions may affect the function of proximal genes, especially when important functional genetic elements are included in the deleted regions. A single gRNA was reported to induce large deletions from a few hundred to several thousand bp in the on-targets in mouse zygotes[11] and in cultured cells[12]. The molecular mechanism underlying the emergence of these large deletions remains unclear. Understanding the molecular mechanisms underlying the emergence of these large deletions should provide much needed information to rationally optimize targeting strategies, so that such events can be avoided or minimized.

We evaluated whether there are large deletions around the on-target site in the *hSOD1* transgene, which was fully targeted in the G1H/Cas9 mice. We used long-range PCR, cloning, and Sanger-sequencing approaches to identify such events. A pair of primers specific to the *hSOD1* transgene was used to amplify a 6.6 kb fragment from a transgene-edited G1H/Cas9 mouse (#8190). The amplified fragment covers over 3 kb on each side of the Cas9 cleavage site (Fig. 5a). The PCR products were cloned into the *pBluescript* II SK(-) plasmid, and individual plasmids were initially analyzed by restriction enzyme digestion.

Among 117 clones, 61 clones showed an insert size similar to wild-type clones (type 1, Wt-like), suggesting no large deletions.

However, 56 clones showed a reduced insert size, suggesting the presence of large deletions of various sizes (Fig. 5a). We sequenced the targeted site of all Wt-like clones and identified 12 small insertions/deletions (Supplementary Fig. 5). The frequencies of the small insertion/deletions in these long-range PCR clones were similar to those in the short-range PCR clones in the same mouse (#8190) (Fig. 3, Supplementary Fig. 5). Enzymatic digestion by restriction endonucleases suggested five types of large deletions, referred to as clone types 2–6, ranging from a few hundred bp to over four kb pairs. We initially sequenced two representative plasmids with 2–3 kb deletions (type 4 and 5 clones) using primers to different regions. Eventually, we determined two deletions, g.1995_4605del2611 and g.1782_5036del3255 for type 4 and 5 clones, respectively (Fig. 5b, c).

In our previously analyzed clones with small insertions/deletions from short-range PCR products, all clones with a deletion over 20 bp showed simultaneous insertions of a few nucleotides in our G1H/Cas9 mice (Fig. 3, Supplementary Fig. 5), a sign of cNHEJ. However, these two large deletions were apparently different from small insertions/deletions in that their end-joining sites were not modified. The exact deletion of 2611 and 3255 bp fragments without any end modifications suggested that these large deletions utilize a molecular mechanism distinct from cNHEJ. On detailed examination of these two large deletions, we found that the deleted DNA included the fragments between two short and proximate identical sequences (PIS), together with one of the PIS (Fig. 5b, c). The sizes of the two PIS were 23 nt and 30 nt, respectively (Fig. 5b, c). The commonality in the deleted regions of these two large deletions suggests that the emergence of the large deletions was highly likely to be mediated by these PIS. We, therefore, searched for PIS shared by both the 5′ and 3′ arms of the Cas9 cleavage site in the entire *hSOD1* gene. We identified one *Alu* element in the 5′ arm, which has multiple PIS in three *Alu* elements in the 3′ arm (Supplementary Fig. 6). We predicted 15 potential large deletions, assuming that a PIS ≥ 15 nt in these *Alu* elements would be able to facilitate the emergence of the large deletions (Supplementary Fig. 6).

Based on this prediction, we directly sequenced our large deletion clones with specific primers, and identified eight interarm deletions (5′ and 3′ arms of Cas9 cleavage site) from 41 clones of types 4–6 (Fig. 5d–j, Table 1). Our restriction enzyme analysis suggested that the deletions in type 2 and type 3 clones are located within the 5′ and 3′ arms, respectively (Fig. 5a). Thus, we searched for PIS within the 5′ or 3′ arms, and found PIS in two *Alu* elements in each arm. We predicted 3 and 11 potential deletions within the 5′ and 3′ arms, respectively (Supplementary Fig. 7). Subsequent sequencing analysis led to the identification of four different intra-arm deletions (Fig. 5j–m). We confirmed seven of these inter- and intra-arm large deletions in a second transgene-edited G1H/Cas9 mouse (#7449) (Table 1).

Altogether, we identified 12 large deletions in a total of 72 clones (Fig. 5n, Table 1). All deletions occurred between two PIS in *Alu* elements, suggesting a mechanism of PIS-mediated intrachromosomal recombination. These deletions occurred not only between the two arms that flank the Cas9 cleavage site, but also within each single arm of the cleavage site (Fig. 5n, Table 1). The interarm deletions could be explained primarily by the single strand annealing (SSA) pathway[29,34]. Whereas the intra-arm deletions have not been previously described to our knowledge, and the pathway underlying these intra-arm deletion events remains unclear.

Notably, the rate of these large deletions was inversely correlated with the distance between two PIS in *Alu* elements, as shown by the facts that we identified 40 clones with a 2.6 kb

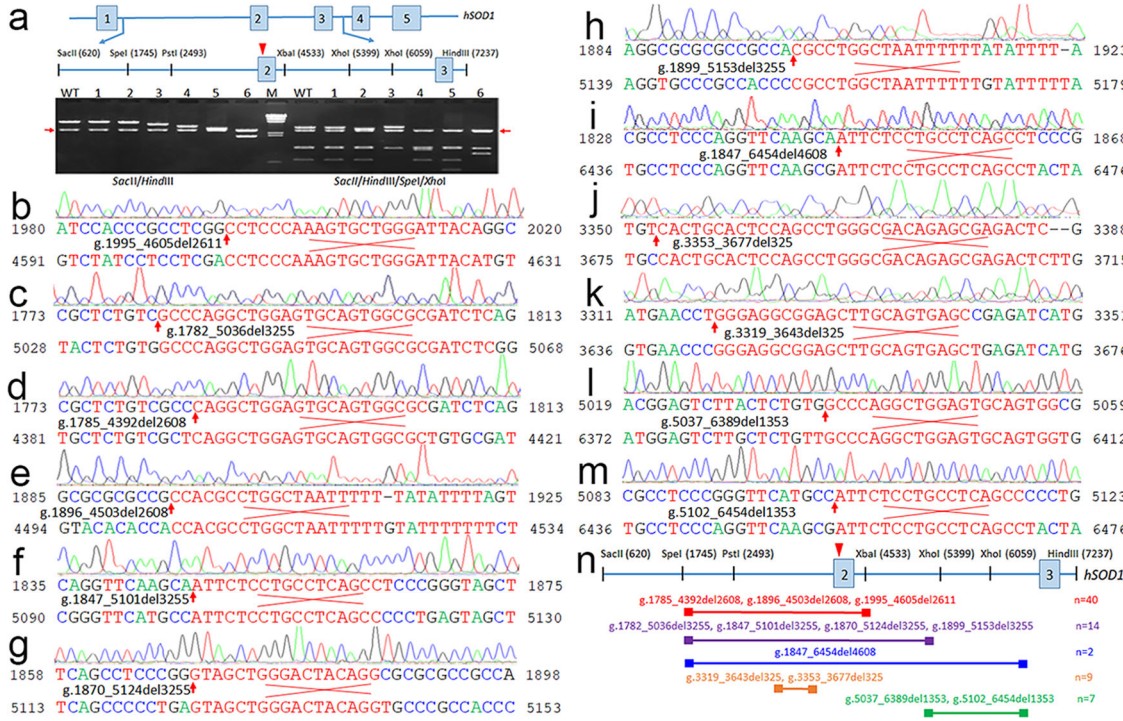

**Fig. 5 PIS-mediated large deletions. a** Schematic representation of the cloning and enzymatic analysis of a 6.6 kb fragment from *hSOD1* in G1H/Cas9 mice. The restriction enzyme sites used for analysis are labeled. Red arrowhead indicates the Cas9 cleavage site. Red arrows indicate plasmid vector DNA. Six types of clones were identified. Type 1 was Wt-like, the other five types showed smaller inserts. **b–m** Twelve deletions were identified, ranging from 325 to 4608 bp in size. Sequencing chromatogram of the DNA sequence around the junction is shown on the top. The PIS is color coded in red. The 5′ junction site is indicated by an arrow. Description of the deletions is based on the NCBI reference sequence (NC_000021.9) for human *SOD1*. **n** Schematic representation of deletions showing the sizes and locations (diagrams are not drawn to scale for clarity). Six *Alu* elements are shown by filled boxes, and the deleted regions are shown by lines. Deletions mediated by different PIS in the same *Alu* elements are shown on the top of the lines. The total number of deletions are summarized on the right, and detailed in Table 1.

**Table 1 Summary of large deletions in the genome edited G1H/Cas9 mice.**

| Deletion | Proximate identical sequence | Size (nt) | Distance (nt) | #8190 | #7449 |
|---|---|---|---|---|---|
| g.1995_4605del2611 | 5′-CCTCCCAAAGTGCTGGGATTACA-3′ | 23 | 2611 | 22 | 7 |
| g.1785_4392del2608 | 5′-CAGGCTGGAGTGCAGTGGCGC-3′ | 21 | 2608 | 5 | 2 |
| g.1896_4503del2608 | 5′-CCACGCCTGGCTAATTTTT-3′ | 19 | 2608 | 3 | 1 |
| g.1782_5036del3255 | 5′-GCCCAGGCTGGAGTGCAGTGGCGCGATCTC-3′ | 30 | 3255 | 6 | 2 |
| g.1847_5101del3255 | 5′-ATTCTCCTGCCTCAGCC-3′ | 17 | 3255 | 1 | 2 |
| g.1870_5124del3255 | 5′-GTAGCTGGGACTACAGG-3′ | 17 | 3255 | 2 | |
| g.1899_5153del3255 | 5′-CGCCTGGCTAATTTTTT-3′ | 17 | 3255 | 1 | |
| g.1847_6454del4608 | 5′-ATTCTCCTGCCTCAGCCT-3′ | 18 | 4608 | 1 | 1 |
| g.3353_3677del325 | 5′-CACTGCACTCCAGCCTGGGCGACAGAGCGAGACTC-3′ | 35 | 325 | 6 | |
| g.3319_3643del325 | 5′-GGGAGGCGGAGCTTGCAGTGAGC-3′ | 23 | 325 | 3 | |
| g.5037_6389del1353 | 5′-GCCCAGGCTGGAGTGCAGTGG-3′ | 21 | 1353 | 4 | 1 |
| g.5102_6454del1353 | 5′-ATTCTCCTGCCTCAGCC-3′ | 17 | 1353 | 2 | |

deletion and 14 clones with a 3.3 kb deletion, but only two clones with a 4.6 kb deletion. Conversely, the rate of these deletions appeared to be positively correlated with the size of PIS in *Alu* elements, as evident by the finding that among 40 clones with a 2.6 kb deletion, the g.1995_4605del2611 with a PIS size of 23nt accounted for approximately three-fourths (29/40) of the total deletion clones; however, g.1785_4392del2608 (PIS size: 21 nt) and g.1896_4503del2608 (PIS size: 19 nt) together comprised about one-fourth (11/40). Similarly, g.1782_5036del3255 (PIS size: 30 nt) accounted for more than half of the total clones with a 3.3 kb deletion (Table 1). Among 12 types of deletion events identified in this study, the smallest PIS was 17 nt (Fig. 5, Supplementary Figs. 6 and 7, and Table 1).

The largest deletion in our G1H/Cas9 mice is 4.6 kb, between two PIS in the most distal *Alu* elements of *hSOD1* (Fig. 5n). Since the key elements mediating the emergence of these deletions appeared to be the short PIS ranging from 17 to 35 bp in the *Alu* elements, it would be interesting to test whether much larger identical (or homologous) sequences could mediate the emergence of even larger deletions. To address this, we analyzed the copy number of *hSOD1* transgene in our genome edited G1H/ Cas9 mice. We previously estimated that G1H mice have ~18 copies of the *hSOD1* transgene, each 11.6 kb in size, which were tandemly integrated into the mouse genome[4]. This estimation was based on our initial analysis using southern blot[4]. Because each transgene is identical (fully homologous) to its neighboring

copies, such a linear incorporation of multiple *hSOD1* transgene copies in G1H mice should allow us to test whether two large identical sequences of 11.6 kb, which are also 11.6 kb apart, can effectively lead to the deletion of the transgene copies. If so, we would expect a minimum of 1~2 *hSOD1* transgene copies left in the genome edited G1H/Cas9 mice.

We employed multiplex ligation-dependent probe amplification (MLPA)[35], a more reliable and accurate method than southern blot to determine transgene copies in our G1H/Cas9 mice. The *hSOD1*-specific probes are outside of the 4.6 kb regions, so that the 12 known large deletions would not affect the copy number assay. We found that control G1H mice have ~30 copies of the *hSOD1* transgene (Fig. 6). Indeed, a loss of variable transgene copies was observed in all five G1H/Cas9 mice that we analyzed, with about 9 to 23 *hSOD1* copies remaining (Fig. 6). The partial loss of the *hSOD1* copies in the transgene-edited G1H/Cas9 mice suggest that the large-sized PIS of 11.6 kb can facilitate the emergence of large deletions, but the efficiency is greatly reduced due to the increased distance between two large PIS. Alternatively, reduction of the copies might also be resulted from the drop of intervening DNA fragments due to simultaneous cleavage by Cas9 at two or more sites in the transgene cluster, as previous studies demonstrated that simultaneous nuclease cleavage at two or more sites on the same chromosome could lead to deletion of the DNA sequences in between, although the rate might be low and appeared to be quite variable[11,36].

## Discussion

Different strategies to decrease mutant SOD1 expression have been previously tested in SOD1-ALS mouse models, and a variety of therapeutic effects were observed[23,37–43]. Some strategies are being tested in clinical trials[44–46]. In the present study, we show that CRISPR/Cas9-mediated gene-editing is an effective strategy to target *hSOD1*, leading to a disease-free condition in two distinct *hSOD1-G93A* transgenic mouse models (G1H and G1L) of ALS. Off-target editing events were observed, but appeared to be rare. Consistent with previous in vitro studies[8], our in vivo data support the hypothesis that the rate of off-target editing events is primarily determined by the "seed sequence" at the 3′ end of the gRNA.

In this study, we employed a transgenic strategy to introduce an *hSOD1*-specific gRNA and CRISPR/Cas9, which are expected to express from an early embryonic stage throughout the lifespan of the mice. This strategy is apparently not applicable to disease treatment in humans. However, due to its maximum editing efficiency, this strategy may offer an advantage in addressing long-term safety concerns in the context of complete editing in preclinical studies in mice. CRISPR/Cas9-mediated double-strand breaks (DSBs) may be repaired through one of four pathways in dividing cells: homologous recombination (HR), cNHEJ, alt-EJ (or MMEJ), and SSA[47]. Nondividing cells, such as mature neurons, lack the HR pathway. Thus, these cells cannot accurately repair Cas9-mediated DSBs through HR. However, these cells still have functional cNHEJ, alt-EJ (or MMEJ), and SSA pathways[48]. Since Cas9-gRNA would keep editing the targets if they were fully repaired though HR, we would not be able to identify any HR-mediated repair events, as shown in our CRISPR/Cas9-targeted mice. Thus, all the DNA repair events identified in our transgene-edited G1H/Cas9 mice, including large deletions, were mediated through cNHEJ, alt-EJ (or MMEJ), and SSA pathways, which are shared by dividing and nondividing cells.

Large deletions previously detected in vitro are a major safety concern[11,12]. This is especially true when functional genetic elements are included in the deleted regions. The previously reported large deletions appeared to be caused by either cNHEJ or alt-EJ with a homology of 1–8 nt in a relatively low frequency[11,12]. Among 148 clones from two G1H/Cas9 mice (#8190 and #7449), we identified 72 clones with large deletions. These deletions accounted for approximately half of all targeted events, suggesting that large deletion events are quite frequent in the human *SOD1* transgene. Detailed examination revealed an important role of the PIS in *Alu* elements in mediating the emergence of these large deletions. The rate of these large deletions appeared to be positively correlated with the size of PIS in *Alu* elements, but inversely correlated with the distance between them.

*Alu* elements are the most abundant short interspersed nuclear elements (SINEs) of ~300 bp that are specific to primates. *Alu* elements account for ~11% of the human genome, with a total of more than one million copies and an average of one *Alu* element/3 kb throughout the human genome[49]. Our data, therefore, suggest that *Alu* elements pose a major risk to large deletions or other rearrangements for CRISPR/Cas9-mediated genome editing, and likely for any other gene editing approaches that utilize double-strand DNA breaks. However, *Alu* elements are not evenly distributed throughout the human genome, providing opportunities to optimize targeting design. For example, in the entire 11.6 kb *hSOD1* transgene, all six *Alu* elements are located in introns 1 and 2, with three *Alu* elements on each side of the Cas9 cleavage site in exon2. If our strategy had been designed to target other exons at a distance from the *Alu* elements, we might not have observed these large deletions. Despite these large deletions, the editing approach appears to be reasonably safe, due to a lack of key genetic elements within the deleted regions.

In addition to SINEs, the human genome contains other types of repetitive sequences, such as long interspersed nuclear elements, LTR retrotransposons, and DNA transposons[49]. Although the copy number of these repeats is less than that of *Alu* elements in the human genome, the size of their individual repeats is much larger. Therefore, these repetitive sequences may also pose a risk for large deletions.

Although large deletions represent a major safety concern for therapeutic development[11,12], our data suggest that these deletions are not random events; rather, they are predictable, as demonstrated in the present study. It is also possible that two or more Cas9 cleavage sites with nearby PIS on different chromosomes may lead to more complex rearrangements, such as chromosome translocations. Accordingly, we suggest that when designing in vivo genome editing strategies with CRISPR/Cas9, repetitive DNA sequences or any PIS ≥15 nt near the Cas9 cleavage sites are best avoided, especially when functional genetic elements are involved and deleterious consequences are predicted. Since the PIS-mediated large deletions/rearrangements are triggered by DSBs, this may also apply to other genome editing approaches where DSBs are involved.

In summary, our data demonstrate a high efficacy of CRISPR/Cas9-mediated in vivo genome editing in the *hSOD1*-ALS mice. Although off-target editing events, large deletions, and other potential rearrangements are challenging safety concerns, they appear to be largely predictable and therefore surmountable by rational selection of the optimal target sites and gRNA sequences. The in vivo data from our long-term study in the *hSOD1*-ALS mice suggest that CRISPR/Cas9 genome editing could be developed as an effective therapeutic approach, with an acceptable risk for treating devastating and uniformly fatal diseases with rapid progression, such as SOD1-A4V linked ALS, which has only 1 year survival after disease onset[2,3].

## Methods

**Development and characterization of transgenic mice**. *hSOD1*-CRISPR/Cas9 (Cas9) transgenic mice were developed using a transgene expressing both SpCas9

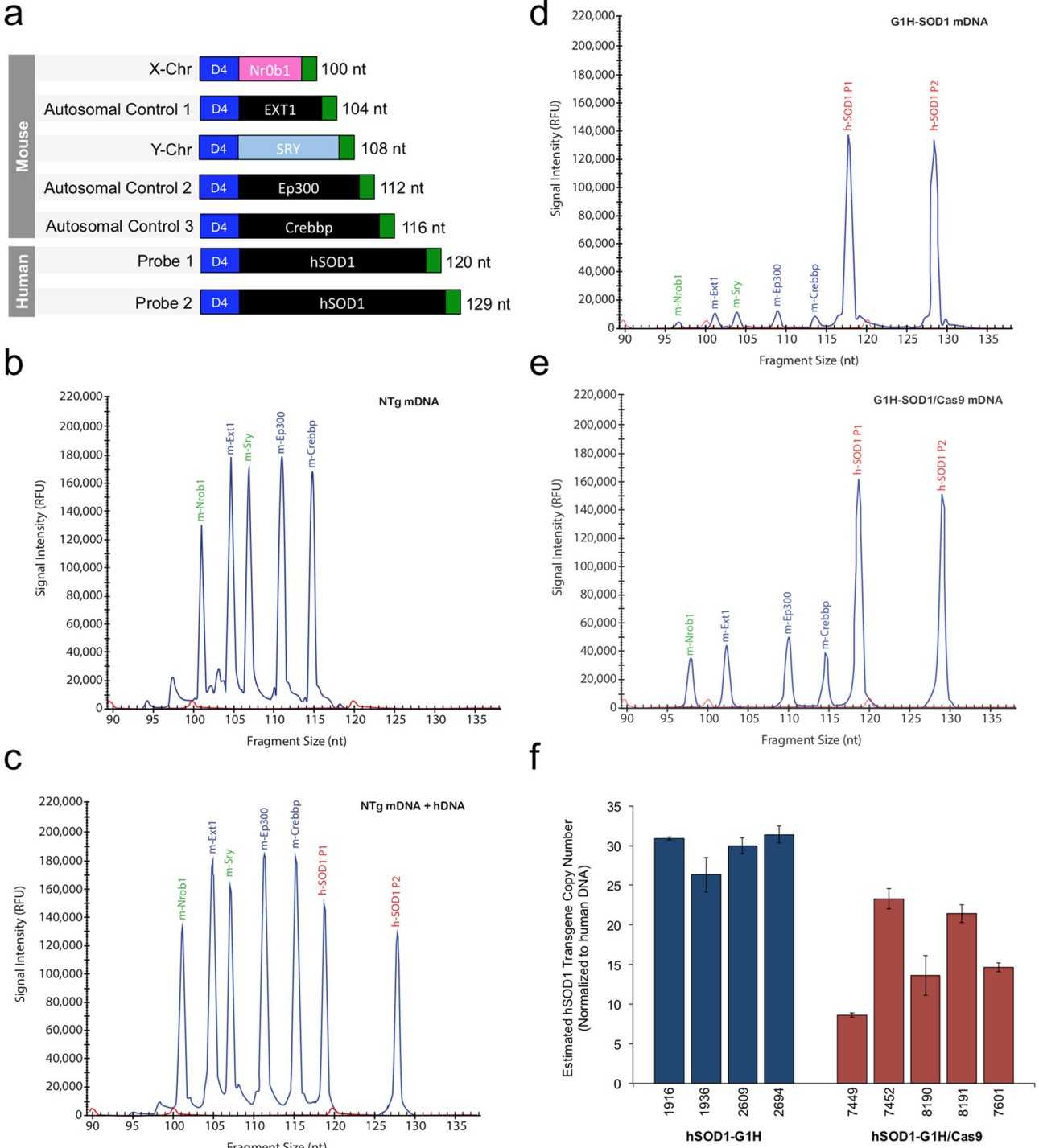

**Fig. 6 Loss of *hSOD1* transgene copy number in G1H/Cas9.** The *hSOD1* transgene copy number was determined by using multiplex ligation-dependent probe amplification (MLPA). **a** Five mouse-specific MLPA probes were designed to hybridize to two sex chromosome-specific (Nr0b1 and SRY) and three autosomal control (Ext1, Ep300, and Crebbp) genes. Two distinct MLPA probes were designed to hybridize to intron 1 (probe 1) and intron 2 (probe 2) of *hSOD1* beyond the known deleted regions. The PCR amplification products generated by each probe differ in size by at least four nucleotides (nt). Nontransgenic mouse DNA (NTg mDNA) in the **b** absence and **c** presence of human DNA (1:1) was used to illustrate *hSOD1* probe specificity. *hSOD1* transgene copy number in the **d** G1H ($n = 4$) and **e** G1H/Cas9 ($n = 5$) mouse lines were estimated by dividing the average peak height of hSOD1 (probe 1) by the average peak height of the three mouse control genes (**e, f**). This ratio was then normalized to the ratio of mouse nontransgenic DNA spiked with human DNA. Three independent MLPA reactions were performed for each condition and the standard deviation (SD) was plotted to illustrate variation. *hSOD1* transgene copies in G1H were estimated to be ~30 (29.7 ± 2.31). The lowest and the highest *hSOD1* transgene copies remained in G1H/Cas9 mice were estimated to be ~9 (8.6 ± 0.25) and 23 (23.3 ± 1.28), respectively.

and an *hSOD1*-gRNA. This gRNA targets to a single cleavage site in exon2 of the human *SOD1* gene (*hSOD1*). In brief, a 20 bp DNA sequence (5′-GTAATG-GACCAGTGAAGGTG-3′) of *hSOD1*-exon2 was inserted into the *Bbs*I site of the plasmid vector, pSpCas9 (BB)-2A-GFP (PX458). The plasmid DNA was digested with *Sac*II and *Not*I, and the 7.5 kb transgene was used for microinjection into mouse fertilized eggs from a *C57BL/6J* inbred strain. Transgenic mice were identified by three sets of PCR. Onset of disease phenotype in mice is defined by showing more than one of the following signs: (i) fine tremor at one or more toes when suspended in the air; (ii) reduced spontaneous movement; (iii) failure to gain weight or loss of weight; (iv) poor grooming; (v) muscle weakness; and (vi) partial paralysis. End stage of the disease is defined as when a mouse is paralyzed and cannot right by itself in 30 s after it is laid on one side or loss 20% of body weight. The animals and animal use protocols have been approved by the Institutional Animal Care and Use Committee of Northwestern University.

**Antibodies**. Three SOD1 antibodies were used in this study: Antibody c-SOD1 was raised with a peptide of the last 28 amino acids that are identical in human and mouse SOD1. This antibody recognizes both human and mouse SOD1[5]. An hs-SOD1 antibody recognizes only human, but not mouse SOD1[5]. To distinguish mouse SOD1 from human SOD1, we generated an ms-Sod1 antibody with a mouse Sod1-specific polypeptide (ASGEPVVLSGQIT). This ms-Sod1 antibody recognizes only mouse, but not human SOD1. Other primary antibodies used in this study included SpCas9 (ab191468, Abcam, Cambridge, UK), GFAP (G4546, Sigma-Aldrich, Inc., St. Louis, MO), IBA1/AIF1 (#MABN92, Millipore Sigma-Aldrich, Inc., St. Louis, MO), β-actin (A5060, Sigma-Aldrich, Inc., St. Louis, MO), α-tubulin (#66031-1-Ig, Proteintech Group, Chicago, IL), ubiquitin (#10201-2-AP, Proteintech Group, Chicago, IL), and ChAT (#AB144P, Millipore Sigma-Aldrich, Inc., St. Louis, MO). The fluorescent secondary antibodies included Alexa Fluor 488 goat anti-mouse IgG (A11029), Alexa Fluor 488 goat anti-rabbit IgG (A11034), Alexa Fluor 555 goat anti-mouse IgG (A21424), and Alexa Fluor 555 goat anti-rabbit IgG (A21429) from Life Technologies/Invitrogen, Grand Island, NY.

**Western blot**. Mouse spinal cords were processed and homogenized. Tissue homogenates were subjected to total protein quantification, gel electrophoresis, and blotted on PVDF membranes. Proteins were detected using antibodies of interest. The membranes were stripped and blotted with an antibody against β-actin or α-tubulin as protein loading controls.

**Off-target prediction, DNA amplification, cloning, and sequencing analysis**. Off-targets were predicted using several computational programs, including http://www.rgenome.net/cas-offinder/[50]; http://crispr.bme.gatech.edu/[51]; http://rgenome.net/cas-designer/[52]; https://cm.jefferson.edu/Off-Spotter[53]; and https://www.idtdna.com/site/order/designtool/index/CRISPR_SEQUENCE. Relaxed Cas9 PAM sequences (NRG, including both NGG and NAG) were used to identify potential off-targets in the mouse genome. The potential off-target sites analyzed in this study included: (i) sites with only 2 nt mismatches (#01-03), (ii) sites with 3 nt mismatches and with a ≥9 nt perfect match in the "seed sequence" adjacent to PAM (#04-#07), (iii) sites with 4 nt mismatches and with a ≥11 nt perfect match in the "seed sequence" adjacent to PAM (#08-#10), (iv) sites with a single DNA bulge (#11, #12), and (v) sites with ≤2 nt mismatch and with a single RNA bulge (#13-#15). A total of 15 potential off-targets were analyzed. PCR primers were designed to cover a genomic DNA fragment > 200 bp flanking the Cas9 cleavage sites. A restriction *Eco*RI site was anchored in the forward primers and a *Hind*III site was anchored in the reverse primers for the purpose of cloning. Two G1H/Cas9 mice (#8190 and #7449) and one G1L/Cas9 mouse (#8306) were analyzed. Briefly, genomic DNA was extracted from the mouse tails by standard methods (69506, QIAGEN, Valencia, CA). PCR amplification protocol consisted of the following steps: incubation at 95 °C for 2 min, 32 cycles of 95 °C (30 s), 58 °C (30 s), and 70 °C (10 min), and a final 5 min extension at 72 °C. The PCR products were purified by using QIAquick PCR purification kit (28104, QIAGEN, Valencia, CA), and digested with restriction enzymes *Eco*RI and *Hind*III. The digested products were purified again with the QIAquick PCR purification kit (Cat. No 28104). The purified products were cloned into CIP-treated *Eco*RI/*Hind*III sites of the plasmid vector *pBluescript* II SK(-). Transfection and plasmid preparation were performed using standard protocols. For DNA Sanger sequencing, fluorescent dye-labeled single-strand DNA was amplified using Beckman Coulter sequencing reagents (GenomeLab DTCS Quick Start Kit) followed by single-pass bidirectional sequencing with a CEQ 8000 Genetic Analysis System (Beckman Coulter, Fullerton, CA).

**Pathology, immunohistochemistry, and confocal microscopy**. Three G1H (#3702, #7455, and #8194), three G1H/Cas9 (#7450, #7452, and #8191), one G1L/Cas9 (#8306), and three wild-type (#7154, #7439, and #8214) mice were included. For the analysis of mouse spinal cords, 9-μm sections were cut from formalin-fixed, paraffin-embedded lumbar spinal cord blocks. The sections were deparaffinized and hydrolyzed. Antigens in the sections were retrieved using a high-pressure decloaking chamber at 121 °C for 20 min[54]. For immunohistochemistry, endogenous peroxidase activity was blocked with 2% hydrogen peroxide. Nonspecific background was blocked with 1% bovine serum albumin. The titers of the

antibodies were determined on the basis of preliminary studies using serial dilution of the antibodies. Biotinylated goat anti-rabbit or anti-mouse IgG were used as the secondary antibodies. Immunoreactive signals were detected with peroxidase-conjugated streptavidin (BioGenex) using 3-amino-9-ethylcarbazole as a chromogen. The slides were counterstained with haematoxylin and sealed with Aqua PolyMount (PolyScience). Quantification of anterior horn spinal cords based on our previously described protocol with minor modifications[55]. Briefly, in every third section, at least ten sections in total from each animal were immunostained with antibody against choline acetyltransferase (#AB144P, Millipore Sigma-Aldrich, Inc., St. Louis, MO). The areas of the anterior horn where motor neurons were counted included laminae VII, VIII, and IX. Cells that met the following criteria in this area were counted as a motor neuron: (i) ChAT positive; (ii) cell body diameter over 10 μm; and (iii) with a clearly defined cytoplasm containing a nucleus. For confocal microscopy, antibodies generated in different species were used in combination. Fluorescence signals were detected with appropriate secondary anti-rabbit, anti-mouse IgG, conjugated with Alexa Fluor 488 or Alexa Fluor 555 using an LSM 510 META laser-scanning confocal microscope with the multitracking setting, and the signals were collected using the band-pass filters. The same pinhole diameter was used to acquire each channel.

**Detection of on-target editing events and large deletions**. Primers hSOD1-cas9TP-1F/hSOD1-cas9TP-1R were used to amplify the human SOD1 *Sac*II/*Hind*III fragment (6617 bp). Primers hSOD1-cas9TP-2F/hSOD1-cas9TP-2R were used to amplify the human 512 bp fragment flanking the Cas9 cleavage site.

hSOD1-Cas9TP-1F: 5′-GCTTGGCCGTGTTCTCGTTCCTGAGGGTCCCGCGG-3′
hSOD1-Cas9TP-1R: 5′-CTACAGAAGCTTCCTATGTTTCACAATG-3′
hSOD1-Cas9TP-2F: 5′-ATACCGCGGTCAGCCTGGGATTTGGACACAGA-3′
hSOD1-Cas9TP-2R: 5′-CAGAAGCTTGAGGATCAATGGAGCCTGGGAG-3′

A restriction *Sac*II site was anchored in forward primers and a *Hind*III site was anchored in the reverse primers for the purpose of cloning.

**Detection of *hSOD1* copy numbers by MLPA**. MLPA was used to ascertain variations in transgene copy number between hSOD1-G1H and hSOD1-G1H/Cas9 expressing mouse lines. Mouse tail DNA (~75 ng per reaction) was extracted using a DNeasy Blood & Tissue kit according to the manufacturer's instructions (69506, QIAGEN Valencia, CA). The synthetic MLPA probe mix consisted of seven unique probes with identical forward (5′-GGGTTCCCTAAGGGTTGGA-3′) and reverse (5′-TCTAGATTGGATCTTGCTGGCAC-3′) primer sequences attached to either a left or right hybridization sequence (LHS and RHS), respectively. The 5′ end of the RHS was phosphorylated to facilitate probe ligation. The LHS and RHS for two sex chromosome-specific (*Nrob1* and *SRY*) and three control (*Ext1*, *Ep300* and *Crebbp*) genes were previously published[35] and then optimized as detailed below. Two MLPA probes were designed to ligate to intron 1 (probe 1) and intron 2 (probe 2) of human *SOD1* (*hSOD1*). The hybridization sequences are as follows:

Nr0b-mouse-LHS: 5′-CTGAGCACATCCGGATGATGCAGAGA-3′
Nr0b-mouse-RHS: 5′-GAGTACCAGATCAGATCCGCTGAACTGAACAG-3′
Ext1-mouse-LHS: 5′-GTACTGTGCGCAGGTGAGTAGCCTTGTTCTCT-3′
Ext1-mouse-RHS: 5′-GCAGCCTCATTGCATACCTCTCCCACACTT-3′
Sry-mouse-LHS: 5′-GGCACAGAGATTGAAGATCCTACACAGAGAGAAA-3′
Sry-mouse-RHS: 5′-TACCCAAACTATAAATATCAGCCTCATCGGAG-3′
Ep300-mouse-LHS: 5′-CAAAAGCCCAATGGCACAGACAGGCTTGACTTCTC-3′
Ep300-mouse-RHS: 5′-CAAACATGGGGATTGGCAGTAGTGGACCAAATCAG-3′
Crebbp-mouse-LHS: 5′-GATCGCCGCATGGAGAACCTGGTTGCCTATGCTAAGA-3′
Crebbp-mouse-RHS: 5′-AAGTGGAGGGAGACATGTATGAGTCTGCTAATAGCAG-3′
hSOD1-1-LHS: 5′-GATGGCGACTGCGCCTGGGCCCGCCTGGTGTCTTC-3′
hSOD1-1-RHS: 5′-GCATCCCTCTCCGCTTTCCGGCTTCAGCGCTCTAGGTCAGGGA-3′
hSOD1-2-LHS: 5′-GCTTTACCGCCTCTGGTCTGGGAGGTGATTGCTCTGCTGCTT-3′
hSOD1-2-RHS: 5′-CCTGTAACTTGCCTGCCTTTCTCCCTGTGTGGGACTCCTGCGGGT-3′

The expected sizes of above seven PCR amplification product are 100, 104, 108, 112, 116120, and 129 nt, respectively. MLPA reaction was performed using MRC-Holland's SALSA MLPA reagent kit (Ek1-Cy5.0) according to the manufacturer's instructions. *hSOD1* transgene copy number was determined by dividing the average peak height of *hSOD1* (probe 1 or probe 2) by the average peak height of three mouse control genes (*Ext1*, *Ep300*, and *Crebbp*). This ratio was then normalized to the ratio of mouse nontransgenic DNA spiked with human DNA (1:1) as an indicator of two endogenous copies of *SOD1*. The estimated copy number generated by *hSOD1* probe 1 and 2 differed by less than 20%.

**Statistical analysis**. Statistical significance of the difference in ChAT-positive neuronal number among G1H, G1H/Cas9, and WT mice was calculated using a one-way ANOVA.

**Reporting summary**. Further information on research design is available in the Nature Research Reporting Summary linked to this article.

## Data availability

All data supporting the findings of this study are available in the main article and its Supplementary Information. All other data are available from the corresponding authors on request.

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

## Acknowledgements

This study was supported by funds from the Foglia Family Foundation, the Les Turner ALS Foundation/Herbert and Florence C. Wenske Foundation Professorship, the Vena E. Schaaf ALS Research Special Trust, the Harold Post Research Professorship in ALS Fund,

the Herbert and Florence Innovative Research Fund, the Les Turner ALS Foundation, and the National Institute of Neurological Disorders and Stroke (NS099623, NS096572, NS094564, NS106307, NS099638, and NS118928). The CRISPR/Cas9 transgenic mice were generated with the assistance of the Northwestern University (NU) Transgenic and Targeted Mutagenesis Laboratory. Imaging work was performed at the NU Center for Advanced Microscopy generously supported by NCI CCSG P30 CA060553 awarded to the Robert H Lurie Comprehensive Cancer Center.

## Author contributions

H.-X.D. and T.S. designed the study. H.-X.D., H.Z., Y.S., G.L., and B.L. performed the mouse phenotype and pathology studies and T.S. verified them. J.L. designed and performed the copy number study. H.-X.D., H.Z., Y.S., G.L. B.L, E.B.R, J.Y., Y.Y., N.Z., Z.Y., and E.L. performed the DNA cloning and sequencing studies. H.-X.D., Y.C.M., and T.S. analyzed the data and wrote the paper. All authors read and approved the final manuscript.

## Competing interests

The authors declare no competing interests.
