## [Peer Review File · Communications Biology]

This manuscript has been previously reviewed at another Nature Research journal. This document only contains reviewer comments and rebuttal letters for versions considered at Communications Biology.

Point-by-point response

We appreciate the reviewers' constructive comments and suggestions, which have greatly helped us to improve this manuscript. We performed new experiments and revised the manuscript to address the reviewers' concerns. Our point-by-point response to each comment is given below.

Reviewer #1 (Remarks to the Author): In this manuscript, Deng and colleagues provided the evidence that the CRISPR/Cas9-based deletion of hSOD1 in SOD1-ALS transgenic mouse models prevented the development of ALS-like phenotypes. In addition, they have shown the preclinical efficacy and long-term safety of CRISPR/Cas9 genome editing. Although their findings are significant to understand the SOD1-mediated ALS pathology and the long-term efficacy and safety of CRISPR/Cas9 genome editing, several major and minor topics should be addressed and improved further as follows.

Specific comments

Reviewer comment 1-1. On page 4 (line 4-5), the authors stated that “It is unknown if large deletions also occur in vivo.”, but Shin et al. (Nature Communications, 2017) has already shown various types of large deletions in vivo using mouse genetics. They have clearly provided the evidence of mosaicism by analyzing over 600 founder mice, which indicate the continuous Cas9 cleavages during the cell divisions (e.g., 2-cell, 4-cell, 8-cell stages and so on) of mouse embryogenesis. Although Deng and colleagues investigated the long-term effect of CRISPR/Cas9-based genome editing, they should not say that they are the first to determine the consequences of CRISPR/Cas9-based genome editing in vivo. Please correct the relevant expression throughout the text.

Response 1-1: We made the requested changes by removing the relevant sentences in pages 4 and 8.

Reviewer comment 1-2. The authors argue that there were rare off-target events in their CRISPR/Cas9-based genome-edited mice, but they only used a single off-target finding software and investigated only a few numbers of potential off-target sites. It has been well recognized that the in silico off-target prediction may not be sufficient to find all potential off-target sites. In

addition, different off-target analysis tools may provide the different results. Indeed, unless performing the whole-genome sequencing analysis, it is hard to conclude that there are rare off-target events. At least, the authors should provide the evidence of rare off-target events by using several different off-target finders or either experimentally prove the off-target effects using GUIDE-seq or Digenome-seq etc.

Response 1-2. We agree with the reviewer that whole-genome sequencing (WGS) would theoretically be a more comprehensive approach to determine the off-target events in our mouse models. However, there are some challenges for WGS to detect off-target events using current WGS platforms due to insufficient depth of coverage and frequent false-positivity. Current WGS platforms have a depth coverage of about 10-40X, and the "ultra-deep" platforms may reach ~100X. Depending on the depth of coverage and/or uneven coverage at different genomic regions, WGS platforms may be limited in their ability to identify mutations with a mutant allele frequency of <5-10% in the mouse genome (Iyer et al, 2018). Several other approaches for detecting off-targets, such as GUIDE-seq, Digenome-seq, CIRCLE-seq, BLISS and DISCOVER-Seq, have been developed. These approaches are very useful to identify off-targets *in vitro* and in cellular models. However, they also have limitations, including producing abundant false positives that need to be further validated, and an inability to operate during *in vivo* editing, especially in our case where two transgenic mouse lines (hSOD1-G93A and hSOD1-CRISPR-Cas9) were cross-bred.

We thank the reviewer for the constructive suggestion regarding the use of several different off-target finders. We updated our analysis to use five different off-target finder programs to predict potential off-targets, including <http://www.rgenome.net/cas-offfinder/>; <http://crispr.bme.gatech.edu/>; <http://rgenome.net/cas-designer/>; <https://cm.jefferson.edu/Off-Spotter/>; https://www.idtdna.com/site/order/designtool/index/CRISPR_SEQUENCE. The predicted potential off-targets are largely overlapping. Our previous analysis did not include potential off-targets with DNA and RNA bulges. We updated our analysis to include five sites with ≤ 2 nt mismatches and with a single DNA or RNA bulge. A total of 2,095 plasmid clones at 15 different potential off-target sites from two G1H/Cas9 mice were individually sequenced using a Sanger sequencing approach. We identified 5 variants in off-target #1 in two mice. We did not observe variants in the other sites. We have updated the methods and related information in the

Supplementary Information under the section “Off-target prediction, DNA amplification, cloning and sequencing analysis”.

Reviewer comment 1-3. The authors have shown that the large deletions are preferentially found near the PIS of Alu elements. Are the PIS-mediated large deletions also found in other previous studies? Many other studies have shown the various large deletions in both in vitro and in vivo and also provided the relevant genome sequences.

Response 1-3. Most previous studies used PCR to amplify relatively short DNA fragments (<600bp) and then sequenced these fragments to determine targeted events. In these studies, deletions induced by microhomology-mediated repair at the junctions have been reported. The length of microhomology in most cases ranges from 2-8bp, depending on the target sites. This short DNA fragment PCR-sequencing approach might fail to identify large deletions. Studies of large DNA deletions caused by a single gRNA have been very limited. Two studies reported CRISPR-Cas9 mediated large deletions at the on-targets, with DNA sequencing data available.

(1) The first report is from Shin et al, who studied the targeted events in mice by injection of a single sgRNA or multiple sgRNAs into fertilized eggs. They observed that simultaneous targeting of the same loci with multiple sgRNAs could generate deletions (up to ~24kb) of the DNA fragments between the Cas9-cleavage sites. No DNA sequence homology was identified at the end-joining sites (Shin et al 2017). They also observed that a single gRNA mostly resulted in small deletions, with a median deletion size of 9 bp. Approximately 20-80% of deletions were found to have microhomology at the end-joining sites (2-5bp). Among 139 founder mice derived from a single gRNA injection, only two “large deletion” (>400bp) events were observed; one was 449bp, and the other was 585bp. These two large deletions did not appear to be related to homologous sequences, since both joining ends lack sequence homology of any size (Shin et al 2017).

(2) The second report is from Kosicki et al, who studied the X-linked *PigA* locus targeted by CRISPR-Cas9 in male mouse ES cells (Kosicki et al 2018). Using long-range PCR and Sanger or PacBio sequencing, they found that deletions over 250bp from the CRISPR-Cas9 cleavage sites could occur in 5-20% of cells at the *PigA*, and 2.6–3.8% of the cells at *Cd9* loci, with the largest deletion of up to 9.5kb. The sequencing data of the deletions at three *PigA* loci (three CRISPR-Cas9 cleavage sites) are available. To examine whether the PIS-mediated large deletions also occurred at these three *PigA* loci in the mouse ES cells, we individually analyzed the DNA

sequences from 190 clones reported by Kosicki et al to determine if there is any homology shared by both ends at each individual end-joining site. The data are summarized in the table below (Reference table 1 for reviewer 1).

Reference table 1 for reviewer 1. Summary of large deletions in PigA by reported Kosicki et al 2018.

Homology at end joining sites (#nt)	0	1	2	3	4	5	6	7	8
Number of clones	13	17	58	47	35	15	0	2	3

Note: “#nt” indicates the number of nucleotides with homology shared by both ends at each individual end-joining site.

Based on our analysis, the number of nucleotides with homology shared by both ends at each individual end-joining site ranges from 1 to 8, suggesting that these deletions were largely mediated by microhomology at the joining ends, with the homology of 2-4nt in length being the most abundant (140/190=73.7%). We have integrated this information into the discussion section in the revised manuscript.

Reviewer comment 1-4. On page 12 (line 13), the author mentioned that the rate of large deletions was low in other previous studies, but Shin et al has shown that over 50 percent of large deletions were found in vivo when simultaneously targeting more than one genomic site with several gRNAs. If Deng and colleagues found that the half of the mutant clones have large deletions, even if they target a single site using one gRNA, then that is unique, but not in the case of simultaneous targeting.

Response 1-4. Yes, we only used a single gRNA in our study, targeting a single site in exon2 of the human SOD1 transgene in the hSOD1-G93A transgenic mice. Large deletions reported by Shin et al were mostly caused by multiple gRNAs. Among 139 founder mice derived from a single gRNA injection, only two “large deletion” (>400bp) events were observed; one was 449bp, and the other was 585bp. These two large deletions did not appear to be related to homologous sequences since both joining ends lack sequence homology of any size (Shin et al 2017).

Reviewer comment 1-5. Regarding figure 2, please provide the deletion sequences found in #7450.

Response 1-5. Unfortunately, since the mouse #7450 (G1H/Cas9) was perfused and fixed with 4% paraformaldehyde for pathological analysis when it was 196 days, fresh DNA from this mouse is unavailable for deletion analysis. However, we performed off-target and deletion analyses of its sibling with the same genotype at the same age (#7449, G1H/Cas9, 196 days). The off-target data from mouse #7449 are included in Fig. 4b and 4c. Among 156 clones analyzed, two mutant clones were identified (delG and delAAGG) in mouse #7449. The deletion sequencing data are summarized in Table 1. Although the number of analyzed clones from #7449 was smaller than that from mouse #8190, the ratio of the clones with different deletions from #7449 was proportional to that from #8190, suggesting that the rate of the large deletions are positively correlated with the size of PIS in *Alu* elements, but inversely correlated with the distance between them. These data are summarized in Table 1.

Reviewer comment 1-6. Have #8190 and #8306 also shown the similar phenotypic features, the inhibition of ALS-like pathology, besides the reduction of Gfap and Aif1 levels?

Response 1-6. Yes, all the G1H mice displayed an ALS-like disease phenotype by 125 days (100-125 days, an average 109.5 ± 6.3 days) and reached end-stage by 155 days (133-155 days, an average of 141.5 ± 8.1 days); whereas, no Cas9-targeted SOD1-G93A/Cas9 mice (n=17; 15 G1H/Cas9, 2 G1L/Cas9, including #8190 and #8306) displayed an ALS-like disease phenotype by the time of sacrifice for pathological and biochemical studies, as shown in Fig. 1d. We performed pathological and biochemical analyses of five Cas9-targeted SOD1-G93A mice (#7450, #7452, #8191, 8190 and 8306). These mice did not develop any ALS-like phenotype by the time of sacrifice for our analyses. We did not observe any ALS-like pathology in the Cas9-targeted mice that we analyzed (such as motor neuron loss, SOD1 aggregation, increased activation of astrocytes and microglia). Therefore, our data suggest that targeting the disease-causing transgene (SOD1-G93A) using CRISPR-Cas9 could effectively prevent the ALS-like phenotype and pathology in the SOD1-G93A mouse models. The phenotype, pathology and biochemical data are summarized in Figs. 1-3, respectively. In addition, we characterized the motor neuron in Cas9-targeted and non-targeted G1H mice, we found that Cas9-targeting prevented the motor neuron loss in the G1H/Cas9 mice. These data are included in the newly added Supplementary Figure 1.

Reviewer comment 1-7. Have you ever analyzed the off-target events in mice at different ages? Even if the off-target events are rare, if there is a chance of continuous generation of unwanted deletions, it would be a critical issue for CRISPR/Cas9-based clinical studies.

Response 1-7. We agree with reviewer 1 that analysis of the off-target events at different ages will provide important information to address whether unwanted off-target events could be continuously generated, and therefore accumulate with age. We analyzed the off-target events using genomic DNA samples isolated from the mouse tails when we sacrificed them. Two G1H/Cas9 mice (#8190 and #7449) were used for analysis of off-target events when they were 196 days (#7449) and 585 days (#8190), respectively. Thus, our data could only show the off-target events at those ages. Among 156 clones from mouse #7449 (196 days), two mutant clones were identified (delG and delAAGG) ($2/156 = 1.28\%$). Among 132 clones from mouse #8190 (585 days), three mutant clones were identified (delG, delAGinsGG and delGTinsGAGTGGTCA) ($3/132 = 2.27\%$).

Indeed, the off-target rate appeared to increase with age (1.28% vs 2.27%; 196 vs 585 days). However, our limited sample size ($n=2$) and the total number of analyzed clones ($156+132 = 288$) might not be sufficient to draw a reliable conclusion. Future studies may be directed to analyze multiple mice at different ages (from weaning (21 days) up to ~30 months) using targeted PCR combined with next generation sequencing strategies, so that a sufficiently large number of individual fragments could be characterized at different ages.

***Minor topics (Reviewer 1):**

Reviewer comment 1-1'. Please clarify whether only one guide RNA was used or several guide RNAs were used to target a single site.

Response 1-1'. We only used a single gRNA to target a single site in exon2 of the human SOD1-G93A transgene in the hSOD1-G93A (G1H and G1L) mice. We clarified this point in the Supplementary Information under the section of “Development and characterization of transgenic mice”.

Reviewer comment 1-2'. In figure 2 (c, f), I assume that Aif1 is shown in green, whereas Gfap is shown in red, but please clarify it in the figure legend.

Response 1-2'. We used an antibody against Gfap (for identification of astrocytes, red) and an antibody against Aif1 (also known as IBA1, for identification of microglia, green) in confocal microscopy. Since the image is very small in panel 2c, the morphology of the cells does not appear to be easily distinguishable. We enlarged a part of the image of panel 2c (lower left corner) to better demonstrate the morphology. Because the astrocytes provide nutrients to neurons, they are often found to make contact with capillaries. In the enlarged image, examples of astrocytes (red, white arrows) surrounding capillaries can be more easily observed (Reference figure for reviewer 1). We revised the figure legend to clarify the labels.

Reference figure 1 for reviewer 1. An enlarged image from panel 2c (lower left corner). The spinal cord sections from an end-stage G1H mouse were analyzed by confocal microscopy using antibodies against Gfap (red) to identify astrocytes and Aif1 (also known as Iba1, green) to identify microglia. Examples of astrocytes surrounding capillaries are indicated (white arrows).

Reviewer comment 1-3'. In figure 2, please indicate how many tissue sections and fields were analyzed for each mouse.

Response #1-3'. We performed pathological analysis using the lumbar spinal cord sections from three G1H mice (#3702, #7455 and #8194) and three G1H/Cas9 mice (#7450, #7452 and #8191). For each mouse, a minimum of 10 sections and 20 fields (anterior horns) were analyzed. In this revised manuscript, we also included a new Supplementary Figure 1 to show motor neuron loss in G1H mice and prevention of such a loss in the G1H/Cas9 mice. We have updated this information in the Supplementary Information under the section “Pathology, Immunohistochemistry and Confocal Microscopy”.

Reviewer comment #1-4'. Please include the line number on each page.

Response #1-4'. It may be a format problem. It seems that we do not have control on that.

Reviewer #2 (Remarks to the Author): Amyotrophic lateral sclerosis (ALS) is an adult-onset motor neuron degeneration disease. Mutations of SOD1 account for about 20% familial ALS. Previous studies have shown that transgenic mice overexpressing mutant human SOD1 develop ALS-like phenotypes, and the disease mechanism is mainly gain of toxicity from mis-folded mutant SOD1, but not the loss of SOD1 function. Therefore, the authors of this manuscript used the CRISPR/Cas9-mediated genome editing to knock out the mutant transgene (human SOD1) in the mouse models to prevent the development of disease phenotypes as the therapeutic approach. They also tested non-specific targeting effect and characterized molecular mechanism of proximate identical sequences-mediated recombination, which may help optimize therapeutic targeting design to avoid or minimize unintended and potentially deleterious recombination events.

Specific comments

Reviewer comment #2-1. The Cas9 and sgRNA were genetically and constitutively expressed in all the cell types from the very beginning. This is hard to link to therapeutic application. First, the mutant gene is knocked out before the disease onset. This doesn't prove knocking out the mutant gene will still be beneficial after the symptoms appear during disease progression. Second, the mutant gene is knocked out during development, probably in cycling cells not in differentiated neurons. The knockout efficiency from this design cannot predict how well this technique could work in post-mitotic neurons when applied in adult nervous system. Third, the DNA repair

mechanism is different in different cell types, especially between mitotic cells (with DNA replication) and post-mitotic differentiated cells (without DNA replication). Therefore, the mechanism of large deletions is possibly different too. Due to all the caveats listed above, it is questionable how genetically introducing Cas9 system can help understand the therapeutic application.

Response #2-1. We agree with the reviewer that in the present study, the Cas9 and sgRNA were genetically and constitutively expressed in all cell types of the mice from the very beginning in our study; while in a clinical setting, the Cas9-sgRNA will be administrated at a certain time point in the human adult life. An ideal strategy to test the therapeutic efficacy and safety would be to introduce the Cas9/sgRNA after the disease onset in our SOD1-ALS (G1H) mice. However, the G1H mice only survive for about one month after the disease onset, which is much shorter than the average survival of ~3 years in ALS patients after the disease onset. One of the major goals in this study is to address potential long-term safety concerns. Two primary issues are involved in the long-term safety: one is duration of the Cas9-sgRNA expression; the other is the efficacy of editing. If the duration of Cas9-sgRNA expression is not long enough, and/or the targeting efficacy is low, the potential long-term risks, such as tumorigenesis, might not be adequately addressed. For example, it took an average of one and a half years for mice with a liver-specific homozygous *p53* deletion to develop hepatocellular carcinoma. Most previous studies reported their safety observations for a few weeks or months after introduction of CRISPR-Cas9. Moreover, their targeting efficiency was generally low, as shown by targeting the central nervous system using AAV-mediated Cas9-gRNA expression (<3%) (Gyorgy et al 2018). We agree that our transgenic approach is not practical for delivering Cas9-sgRNA for therapeutic purposes. However, no ideal approaches are currently available to effectively deliver Cas9-sgRNA and achieve complete targeting/editing efficiency. As a preclinical strategy, the transgenic approach offers certain advantages over the other delivery approaches, such as AAV and nanoparticles, to address some potential long-term safety concerns due to the continuous expression of Cas9-gRNA throughout entire lifespan of the mice (more than 2 years, up to 32 months in G1H-Cas9 mice) and full targeting/editing efficiency. Thus, although this approach is not applicable for therapeutic gene editing, it allows us to monitor the phenotype and pathology in our ALS mouse models, in which a maximum editing efficiency can be achieved, and Cas9-gRNA expression can be maintained for

a long period of time. We believe this is important for a preclinical study, especially where Cas9-induced tumorigenesis is concerned.

We also agree with the reviewer that “the DNA repair mechanism is different in different cell types, especially between mitotic cells (with DNA replication) and post-mitotic differentiated cells (without DNA replication)”. Cas9-mediated double-strand breaks (DSBs) may be repaired through one of the three mechanisms in dividing cells: non-homologous end joining (NHEJ), microhomology-mediated end joining (MMEJ), and homologous recombination (HR). Indeed, mature neurons are non-dividing cells, which lack HR mechanism; thus mature neurons cannot accurately repair Cas9-mediated DSBs through HR. However, mature neurons still have functional NHEJ and MMEJ pathways. Since Cas9-sgRNA will keep cutting the targets and potential off-targets if they are fully repaired through HR mechanism in the mice, we would not be able to identify any HR-mediated repair events in the mice. As shown in Figs 3-5 and Supplementary Figs 1-3, the identified DNA editing events in our study appeared to be repaired exclusively through NHEJ and MMEJ pathways, which are shared by dividing cells and non-dividing neurons. Indeed, we found *Alu*-derived PIS-mediated large deletions. These deletions could be classified as DNA repair events through MMEJ pathways, more specifically through a single strand annealing (SSA) mechanism, as outlined in the discussion section of the revised manuscript.

Reviewer comment #2-2. The non-specific targeting is identified by candidate approach, not whole genome sequencing. There might be mutations in other regions as well.

Response #2-2. We agree with the reviewer that whole-genome sequencing (WGS) would theoretically be a more comprehensive approach to determine the off-target events in our mouse models. However, there are some challenges for WGS to detect off-target events using current WGS platforms due to insufficient depth of coverage and frequent false-positivity. Current WGS platforms have a depth coverage of about 10-40X, and the "ultra-deep" platforms may reach ~100X. Depending on the depth of coverage and/or uneven coverage at different genomic regions, WGS platforms may be limited in their ability to identify mutations with a mutant allele frequency of <5-10% in the mouse genome (Iyer et al, 2018). Since WGS also yields false-positivity, the potential variants identified by WGS need to be further validated using targeted sequencing approaches. Silico prediction is a current standard approach employed to reliably identify potential

off-target events, with subsequent validation using specific targeted PCR-amplification followed by cloning or high-throughput sequencing platforms (Sun et al Science Advances 2020). We have now used five different off-target finder programs to predict potential off-targets. The predicted potential off-targets are largely overlapping. Our previous analysis did not include potential off-targets with a DNA or RNA bulge. We updated our analysis to include five sites with ≤ 2 nt mismatches and with a single DNA or RNA bulge. Overall, a total of 2,095 plasmid clones at 15 different potential off-target sites from two G1H/Cas9 mice were individually sequenced using a Sanger sequencing approach. We identified 5 variants in off-target #1 in two mice. We did not observe variants in the other sites. We have updated the methods and related information in the Supplementary Information under the section “Off-target prediction, DNA amplification, cloning and sequencing analysis”.

Reviewer comment #2-3. What age of mice were characterized for the pathology phenotypes in Figure 2? Lack of motor neuron quantification. Lack of motor behavior measurement.

Response #2-3. The SOD1-G93A transgenic mice were developed by our group 26 years ago (Gurney et al. 1994). These mice have displayed a very consistent disease course over the past two decades since they were developed, with a narrow window for the disease onset and survival. These mice have been the most widely used for ALS pathogenic studies and therapeutic testing. The disease onset of G1H (SOD1-G93A high expression line) on the *C57BL/6J* genetic background occurred by 3-4 months, and a survival of 4-5 months. The G1H mouse (#3702) with pathology shown in Fig. 2 was 148 days, and the G1H/Cas9 mouse (#7450) was 196 days. The G1H/Cas9 mouse (#7450) was more than one month older than G1H mouse (#3702) because we wanted to test whether the Cas9-targeted mice would develop any signs of ALS by ~200 days. To address the reviewer’s concern regarding the phenotype of the mice, we have added a paragraph in the “Development and characterization of transgenic mice” section in the supplementary information. It includes parameters that we routinely used to define the onset of disease and end-stage of the disease in our laboratory. The overall data were summarized in Fig. 1d.

To address the reviewer’s concern about motor neuron quantification, we analyzed spinal cord sections from three G1H (#3702, #7455 and #8194), three G1H/Cas9 (#7450, #7452 and #8191)

and three wild-type (#7154, #7439 and #8214) mice. We performed immunohistochemistry using lumbar spinal cord sections, which were stained with an antibody against choline acetyltransferase (ChAT). We quantified motor neurons in the anterior horns of each spinal cord section. We analyzed every third section, at least 10 sections total, from each animal. The areas of the anterior horn where motor neurons were counted included laminae VII, VIII and IX. Cells that met the following criteria in this area were counted as a motor neuron: (i) ChAT positive; (ii) cell body diameter over 10 μm ; (iii) with a clearly defined cytoplasm containing a nucleus. These data are summarized in Supplementary Figure 1.

Supplementary Figure 1. Prevention of motor neuron loss by CRISPR-Cas9 targeting in G1H/Cas9 mice. Lumbar spinal cord sections were stained with an antibody against choline acetyltransferase (ChAT) by immunohistochemistry. Whole cross-section images are shown for G1H (a), G1H/Cas9 (b) and wild-type (WT, c) mice. Representative ChAT-positive motor neurons are indicated by arrows. Scale bar, 200 μm . Shrinkage of the spinal cord, together with the shrinkage and loss of ChAT-positive neurons in the anterior horns of the G1H mice are shown in panel (a). (d) ChAT-positive neurons in the anterior horns from each of three genotypes were counted. The average number of ChAT-positive neurons/section in the anterior horns from G1H (34 sections), G1H/Cas9 (33 sections) and WT (41 sections) is 11.94 ± 2.60 , 22.8 ± 6.76 and 23.05 ± 6.38 , respectively. *** indicates $p < 0.0001$.

Consistent with previous data from us and others, G1H mice lost about half of the motor neurons by end-stage, while such a loss of motor neurons was prevented by Cas9-targeting in the G1H/Cas9 mice. We have included detailed information in the “Pathology, Immunohistochemistry and Confocal Microscopy” section in the Supplementary Information.

REVIEWERS' COMMENTS:

Reviewer #1 (Remarks to the Author):

The authors provided detailed explanation for all the points that the reviewers raised. But the writing of the manuscript need to be improved. Some of the rebuttal should be included in the manuscript.

For example:

Response 2-1: The differences between dividing cells and post-mitotic neurons need to be included in discussion. The caveats and limitation of the strategy need to be discussed. The DNA repair pathways that could be shared and potentially account for the large deletions should be discussed, with proper citations.

Response 2-3: The time line and detailed time points of each experiment should be included in the figure legend. Again, the authors provided detailed information in the rebuttal, but the information was not included in the manuscript.

Response 1-7: how the off-target events changes during aging should be included in result and discussion.

Reviewer #2 (Remarks to the Author):

The authors have done a satisfactory job of addressing the reviewer comments. My major criticism with this manuscript version of the manuscript is that it does not effectively contextualize the author's findings in the context of the therapeutic gene-editing and gene silencing landscape for SOD1-ALS. The authors should cite and elaborate in their discussion a recent study demonstrating that CRISPR base editing can be used disrupt SOD1 in adult G93A-SOD1 mice to slow disease progression (Lim et al. Mol. Ther. 2010), which is an advance over the first application of CRISPR editing for ALS, and two recent clinical reports documenting the effects of administering a SOD1-targeting ASO and a miRNA to ALS patients (Mueller et al. N. Eng. J. Med. 2020 and Miller et al. N. Eng. J. Med 2020, respectively). The authors in fact should explicitly mention that their report does not represent a therapeutic strategy, as there is no targeted delivery to affected cells, and instead highlight that it is a step toward determining the long-term of gene-editing approach (the latter of which they do specify).

Point-by-point response

We appreciate the reviewers' comments and suggestions. We have made the requested changes accordingly. Our point-by-point response to each comment is given below.

REVIEWERS' COMMENTS:

Reviewer #1 (Remarks to the Author): The authors provided detailed explanation for all the points that the reviewers raised. But the writing of the manuscript needs to be improved. Some of the rebuttal should be included in the manuscript.

Reviewer comment #1-1: For example: Response 2-1: The differences between dividing cells and post-mitotic neurons need to be included in discussion. The caveats and limitation of the strategy need to be discussed. The DNA repair pathways that could be shared and potentially account for the large deletions should be discussed, with proper citations.

Response #1-1. We have included the following paragraph in the discussion section: "In this study, we employed a transgenic strategy to introduce a *hSOD1*-specific gRNA and CRISPR/Cas9, which are expected to express from an early embryonic stage throughout the lifespan of the mice. This strategy is apparently not applicable to disease treatment in humans. However, due to its maximum editing efficiency, this strategy may offer an advantage in addressing long-term safety concerns in the context of complete editing in preclinical studies in mice. CRISPR/Cas9-mediated double-strand breaks (DSBs) may be repaired through one of four pathways in dividing cells: homologous recombination (HR), cNHEJ, alt-EJ (or MMEJ) and SSA⁴⁷. Non-dividing cells, such as mature neurons, lack HR pathway. Thus, these cells cannot accurately repair Cas9-mediated DSBs through HR. However, these cells still have functional cNHEJ, alt-EJ (or MMEJ) and SSA pathways⁴⁸. Since Cas9-gRNA would keep editing the targets if they were fully repaired through HR, we would not be able to identify any HR-mediated repair events, as shown in our CRISPR/Cas9-targeted mice. Thus, all the DNA repair events identified in our transgene-edited G1H/Cas9 mice, including large deletions, were mediated through cNHEJ, alt-EJ (or MMEJ) and SSA pathways, which are shared by dividing and non-dividing cells."

Reviewer comment #1-2: Response 2-3: The time line and detailed time points of each experiment should be included in the figure legend. Again, the authors provided detailed information in the rebuttal, but the information was not included in the manuscript.

Response #1-2: We revised the figure legends accordingly.

In Fig. 2, we added the following age information into the figure legend: “Representative images from an end stage G1H (#3702, 148 days) and *hSOD1*-targeted G1H/Cas9 mice (#7450, 196 days) are shown.”

In Fig. 3, we added the following age information into the figure legend: “Efficient targeting of *hSOD1* in the G1H/Cas9 mice. (a) Targeting events identified in a G1H/Cas9 mouse (#8190, 585 days).”

In Fig. 4, we added the following age information into the figure legend: “Two G1H/Cas9 mice were analyzed (#8190, 585 days: a; #7449, 196 days: b).”

Reviewer comment #1-3: Response 1-7: how the off-target events changes during aging should be included in result and discussion.

Response #1-3: We added the following paragraph at the end of “Off-target editing events” section in the results: “We analyzed off target events in two mice at ages of 196 days (#7449) and 585 days (#8190), respectively. The off-target rate appeared to increase with an increased time period of CRISPR/Cas9 in the mice [1.28% (2/156) at 196 days vs 2.27% (3/156) at 585 days]. However, our limited sample size (n=2) and total number of analyzed clones (156+132 = 288) might not be sufficient to draw a reliable conclusion. Future studies may be directed to analyze multiple mice at different ages using targeted PCR combined with next generation sequencing strategies, so that a sufficient number of individual fragments could be characterized at different ages.”

Reviewer #2 (Remarks to the Author): The authors have done a satisfactory job of addressing the reviewer comments. My major criticism with this manuscript version of the manuscript is that it does not effectively contextualize the author's findings in the context of the therapeutic gene-editing and gene silencing landscape for SOD1-ALS. The authors should cite and elaborate in their discussion a recent study demonstrating that CRISPR base editing can be used disrupt SOD1 in

adult G93A-SOD1 mice to slow disease progression (Lim et al. Mol. Ther. 2010), which is an advance over the first application of CRISPR editing for ALS, and two recent clinical reports documenting the effects of administering a SOD1-targeting ASO and a miRNA to ALS patients (Mueller et al. N. Eng. J. Med. 2020 and Miller et al. N. Eng. J. Med 2020, respectively). The authors in fact should explicitly mention that their report does not represent a therapeutic strategy, as there is no targeted delivery to affected cells, and instead highlight that it is a step toward determining the long-term of gene-editing approach (the latter of which they do specify).

Response to comments of reviewer #2. We have added three references mentioned by reviewer #2 in the first paragraph of the discussion section. We have also clearly discussed that the transgenic strategy does not represent a therapeutic strategy, but it is a step toward determining the long-term safety of the CRISPR/Cas9 gene-editing approach. We have included the following sentences in the discussion section (second paragraph): “In this study, we employed a transgenic strategy to introduce a *hSOD1*-specific gRNA and CRISPR/Cas9, which are expected to express from an early embryonic stage throughout the lifespan of the mice. This strategy is apparently not applicable to disease treatment in humans. However, due to its maximum editing efficiency, this strategy may offer an advantage in addressing long-term safety concerns in the context of complete editing in preclinical studies in mice.”